# *Arthrobotrys musiformis* (Orbiliales) Kills *Haemonchus contortus* Infective Larvae (Trichostronylidae) through Its Predatory Activity and Its Fungal Culture Filtrates

**DOI:** 10.3390/pathogens11101068

**Published:** 2022-09-20

**Authors:** Gustavo Pérez-Anzúrez, Agustín Olmedo-Juárez, Elke von-Son de Fernex, Miguel Ángel Alonso-Díaz, Edgar Jesús Delgado-Núñez, María Eugenia López-Arellano, Manasés González-Cortázar, Alejandro Zamilpa, Ana Yuridia Ocampo-Gutierrez, Adolfo Paz-Silva, Pedro Mendoza-de Gives

**Affiliations:** 1Laboratory of Helminthology, National Centre for Disciplinary Research in Animal Health and Innocuity (CENID-SAI), National Institute for Research in Forestry, Agriculture and Livestock, INIFAP-SADER, Morelos, Jiutepec CP 62550, Mexico; 2Production Sciences and Animal Health, Faculty of Veterinary Medicine and Zootechnics, National Autonomous University of Mexico, Coyoacán CP 04510, Mexico; 3Tropical Livestock Center, Faculty of Veterinary Medicine and Zootechnics, National Autonomous University of Mexico, Martínez de la Torre CP 93600, Mexico; 4Faculty of Agricultural, Livestock and Environmental Sciences, Autonomous University of the State of Guerrero, Iguala de la Independencia CP 40040, Mexico; 5South Biomedical Research Center, Social Security Mexican Institute (CIBIS-IMSS), Xochitepec CP 62790, Mexico; 6Department of Animal Pathology, Faculty of Veterinary, University of Santiago de Compostela, 27142 Lugo, Spain

**Keywords:** nematophagous fungi, *Arthrobotrys*, nematodes predation, biocontrol, natural compounds

## Abstract

*Haemonchus contortus* (Hc) is a parasite affecting small ruminants worldwide. *Arthrobotrys musiformis* (Am) is a nematode-trapping fungi that captures, destroys and feeds on nematodes. This study assessed the predatory activity (PA) and nematocidal activity (NA) of liquid culture filtrates (LCF) of Am against Hc infective larvae (L3), and additionally, the mycochemical profile (MP) was performed. Fungal identification was achieved by traditional and molecular procedures. The PA of Am against HcL3 was performed in water agar plates. Means of non-predated larvae were recorded and compared with a control group without fungi. LCF/HcL3 interaction was performed using micro-tittering plates. Two media, Czapek–Dox broth (CDB) and sweet potato dextrose broth (SPDB) and three concentrations, were assessed. Lectures were performed after 48 h interaction. The means of alive and dead larvae were recorded and compared with proper negative controls. The PA assessment revealed 71.54% larval reduction (*p* < 0.01). The highest NA of LCF was found in CDB: 93.42, 73.02 and 51.61%, at 100, 50 and 25 mg/mL, respectively (*p* < 0.05). Alkaloids and saponins were identified in both media; meanwhile, coumarins were only identified in CDB. The NA was only found in CDB, but not in SPDB. Coumarins could be responsible for the NA.

## 1. Introduction

Soil-born nematodes include a wide variety of parasites affecting nematodes of importance in agriculture and the livestock industry [1,2,3]. *Haemonchus contortus* is one of the most pathogenic parasites responsible for significant damage to the health and productive potential of small ruminants [4,5]. This nematode is responsible for severe anaemia, hypoproteinaemia, oedema and diarrhoea, which can lead to death in young animals [6,7]. This and other genera/species of parasitic nematodes belonging to the group of gastrointestinal parasitic nematodes have an important economic impact in many countries. For example, in Mexico, a study of the economic losses caused by gastrointestinal parasitic nematodes in cattle revealed an estimated loss of USD 445 million per year [8]. In India, USD 103 million losses have been attributed to the pathogenic effect of *H. contortus* on sheep and goats [9]. The most common way to control these parasites has been based on the continuous use of chemical anthelmintic drugs synthesised in the laboratory. These drugs help to reduce the parasitic burden on these animals in some way; however, there are a number of disadvantages associated with their use, i.e., the presence of anthelmintic resistance in the parasites [10,11], the potential contamination of meat or milk or derivates for human consumption [12] and the contamination of soil and aquifers that can have detrimental environmental effects [13]. Such negative findings have promoted a poor reputation for their use. Nematophagous fungi (NF) are soil microorganisms living as saprophytes, which possess a unique characteristic consisting of their ability to modify their physiological and metabolic behaviour and the ability to transform their mycelia in trapping devices that are particularly designed to capture and destroy nematodes, changing from saprophytic organisms to predators or parasites of nematodes [14,15]. The predatory activity of NF is complemented by the production of several enzymes and metabolites with nematocidal activity [16]. *Arthrobotrys musiformis* is a NF species that has been evaluated in a few studies, mainly focused on its predatory activity [17,18]. The process of degradation of nematodes by *A. musiformis* has been attributed to a proteolytic enzyme mechanism [19]. Recent studies have reported that one strain isolated from Taiwan, *A. musiformis*, is able to produce several small peptides with nematocidal activity that have been found in the predatory stage of this species [20]. The production of bioactive compounds by NF could be a promising source of potential biotechnological tools for the control of gastrointestinal parasitic nematodes. The objectives of the present study were: (1) to assess the in vitro predatory activity of *A. musiformis* against *H. contortus* infective larvae; (2) to assess the nematocidal effect of fungal liquid culture filtrates (FCF) against same stage of the parasite, and (3) to identify the mycochemical compound profile.

## 2. Results

### 2.1. Traditional Taxonomy by Morphological Identification

After 15 days of incubation, the microscopic observation of the water agar plate surfaces sprinkled with soil samples revealed the presence of fungal aerial structures consisting of the formation of tall and erect conidiophores with the presence of apical conidia clusters. Conidia appeared elongate-obovoidal and slightly curved and separated by a septum. Trapping devices consisted of a three-dimensional adhesive net. Sparse chlamydospores were seen in old cultures (Figure 1).

The measurements of morphological structures of taxonomic importance, including conidia length and width, conidiophore length, as well as the presence of chlamydospores and type of trapping devices, are shown in Table 1.

The different morphological characteristics observed, as well as the measurements of structures of taxonomic importance, suggested that the genus and species of NF isolated corresponded to *A*. *musiformis*, according to the taxonomic keys mentioned in the Materials and Method section.

### 2.2. Molecular Taxonomy

A comparison of the 18S region, internal transcript spacer 1, the 5.8S ribosomal region, the internal transcript spacer 2 of the complete sequence, and a partial 28S ribosomal sequence was performed by aligning the sequences on the Blast tool of NCBI, and revealed 99.83% similarity with *A. musiformis* (Table 2).

The molecular analysis of fungal DNA sequences followed by aligning and the high similarity with other isolates reported at the NCBI, led to the conclusion that our isolate corresponded to the NF *A.*
*musiformis,* confirming the results of the traditional morphological analysis. The phenogram obtained by the UPGMA algorithm using the Jukes–Cantor nucleotide distance measure and bootstrap analysis with 1000 replicates, is shown in Figure 2.

### 2.3. Predatory Activity

Photographic evidence of the predatory activity of *A. musiformis* against *H. contortus* larvae is shown in Figure 3.

The results of the fungal predatory activity assay including the group with larvae and the fungus (group 1), and the group with only larvae (group 2), as well as the means of recovered *H. contortus* larvae after the fungus/nematode interaction and the larval reduction percentage attributed to the predatory activity, are summarised in Table 3.

### 2.4. Nematocidal Activity of the Culture Filtrates from A. musiformis against H. contortus Infective Larvae

The results of the FCF/larvae confrontation at three different concentrations including the dead and total larvae recovered and the mortality percentages, are summarised in Table 4.

### 2.5. Microscopical Findings

The main morphological changes identified in *H. contortus* infective larvae after exposure to the fungal culture filtrate are shown in Figure 4. The major damages in larvae after exposure to the FCF were seen in the anterior extreme of the larvae. Most larvae appeared with a widening and malformations characterised by swelling and the rugosity of a cuticular surface with the loss of internal organ cell architecture.

### 2.6. Mycochemical Group Identification Profile

The qualitative analysis of chemical myco-constituent groups in FCF, using the proper chemical reagents, are summarised in Table 5. In Czapek–Dox broth medium (CDB), the analysis revealed the presence of alkaloids, coumarins and saponins; meanwhile, in Sweet Potato Dextrose Broth (SPDB), alkaloids and saponins were identified with a positive reaction.

## 3. Discussion

### 3.1. Traditional Taxonomy

At first sight, the morphological details of the fungal taxonomical characteristics under the microscope, i.e., the type, length and shape of conidiophores and conidia, suggested to us the presence of *A. musiformis* or perhaps *A. dactyloides*; however, the presence of three-dimensional adhesive nets produced by our isolate was an overwhelming feature of our isolate that led us to discard the species *A. dactyloides,* which produces constricting rings [21]. After this finding, we immediately built up an idea about the genus/species of fungus we were working with. The presence of a single, not branched, long and erect conidiophore, bearing an apical conidia cluster arranged upwards, most of them with four to six conidia per cluster and two-cell elongated–obovoidal conidia with the presence of a middle or sub-middle septum, suggested that we had probably isolated an *A. musiformis* strain [22]. Nevertheless, it is important to remark that other genera/species of NF with similar characteristics have recently been described. For example, *Arthrobotrys*
*eryuanensis* has similar conidia in terms of shape and size; in fact, both species share almost the same conidia size: *A. eryuanensis* conidia measure from 18–44.5 × 5–11.5 μm; meanwhile the dimensions of *A. musiformis* conidia range from 20–47.5 × 7–12.5 μm. Therefore, the measurements between both species can overlap. On the other hand, with respect to conidiophores, *A. eryuanensis* produces two types of conidia, macro- and microconidia, with the first having one septum and are partly curved and partly symmetrical and also possess microconidia that are aseptate and truncate at the base with papillate bulge. In contrast, *A. musiformis* produces only one type of conidia (macroconidia). Similarly, *A. eryuanensis* produces branched conidiophores, while *A. musiformis* produces only unbranched conidiophores, in contrast [23]. 

### 3.2. Molecular Taxonomy

After aligning the obtained sequence of our isolate, a high similarity percentage was found with other isolates reported at the NCBI, particularly with *Arthrobotrys* sp. FZ-2020b strain ZB129 (GenBank code, MT612105.1) which has recently been reported as *A. eryuanensis* [23]. Such high similarity of *A. eryuanensis* with our isolate was determined after considering some phenotypic features, mainly in the difference of unbranched conidiophores and the presence of only macroconidia in *A. musiformis*, which was in contrast to *A. eryuanensis* that produces branched conidiophores and macro- and microconidia. There were other strains including the isolate code MH855842.1 and the isolate code KP859624.1 strain BCRC 32758, reported by Tzean et al. (2016) [24]. Both isolates were reported as *A. musiformis* at GenBank. Other isolates recorded at GenBank with slightly lower similarities were also found. However, a phylogenetic tree showed that our isolate was more closely related to the *A. musiformis* isolate, reported by Liou and Shean (1997) [25], than with *A. eryuanensis* (MT612105.1), which supported our findings in traditional taxonomy.

### 3.3. Predatory Activity

The in vitro predatory activity percentage obtained with our *A. musiformis* isolate against *H. contortus* infective larvae was similar to other NF. In general, the predatory activity of this kind of fungi ranges between 60 and 95%. Some results regarding the in vitro predatory activity of *A. musiformis* against taxonomic different genera/species of nematodes are summarised in Table 6. In this Table, we can observe *A. musiformis* has been assessed against different kinds of nematodes, including nematodes of importance for agriculture and the livestock industry, specifically against ruminant parasitic nematodes and against predatory nematodes and free-living nematodes. A predatory activity higher than 70% is a very good activity; if we consider that, using NF for the control of animal parasitic nematodes should be considered only as a tool of control together with other control measures, or even using a combined method with two or more fungal genera/species to achieve a more effective control [26]. It is important to consider that the use of nematode natural antagonists is only part of an integrated control system that can involve different strategies of control [27]; for example, the use of a high protein and energy-based diet, which promotes immune self-defence mechanisms [28,29], grazing management [30,31], vaccines [32,33] and the use of plants/plant metabolites [34,35].

### 3.4. Nematocidal Activity of Fungal Culture Filtrates

The fact that NF are able to develop trapping devices when nematodes are close to them, has been well documented. This characteristic has been deeply studied, and the reason why fungi are stimulated to modify their mycelia in specialised organs to capture and destroy nematodes has been determined [17]. This change in physiology and the nutritional behaviour of NF has been attributed to a cuticle peeling structure of protein nature, called “nemin” [43]. Nematophagous fungi have developed a nemin-recognising receptor system in their cellular surface; this system allows the nemin particles to bind the fungal receptors and, as soon as this binding happens, fungus is stimulated to initiate the morphogenesis process, which implies the transformation of mycelia in traps [44,45]. However, this process is only one part of the mechanisms strategically used by this group of fungi to eventually feed on prey nematodes. Other strategies contribute to the process of attraction, adhesion and cuticular degradation, penetration, and the invasion of nematode bodies, as well as eventually the fungal nutrition from the nematode internal tissues [46]. These mechanisms are developed in a sequence of biological, physiological and biochemical steps as follows: (a) production of attractants molecules that mimic sexual and food olfactory clues to lure nematodes [47]; (b) production of adhesive extracellular polymers that attach to the nematode cuticular surface contributing to the trapping process [48]; (c) nematode paralysing substances or nematotoxins [49]; (d) an enzymic system specially directed to degrade the nematode cuticle components to traverse and penetrate the cuticular wall [50,51]; and (e) production of nematocidal metabolites [16,52,53]. The results obtained in the present study showed differences in the nematocidal activity of FCF of *A. musiformis* growing in different media. The highest nematocidal activity of FCF was seen in the fungus growing in CDB at the three assessed concentrations; meanwhile, FCF of *A. musiformis* growing in SPDB medium showed a very low nematocidal activity at the three concentrations (16.91–26.80). A concentration-dependent effect was observed, with the highest concentration (100 mg/mL) being the one that produced the highest larval mortality (>93%). Nevertheless, the 50 mg/mL concentration resulted in an important mortality (>70%) that is still a very good efficacy, mainly if we consider that we only assessed a simple FCF. It is important to consider that a FCF is only a crude extract and possesses a large number of myco-compounds. At this point, we do not know which compound or compounds were responsible for such activity. It is important to remark that the present study was part of a wider project focused on obtaining nematocidal bio-compounds from NF with potential use against GIN in ruminants. The authors of the present study will continue with the chromatographic process of these filtrates through a bio-guided study to identify the molecules responsible for the nematocidal activity. 

### 3.5. Photographic Analysis

The analysis of changes evidenced by microscopy showed important damage caused by FCF of *A. musiformis* against *H. contortus* infective larvae; not only at the cuticle level, but also to cells of the internal organs. These changes were specifically severe at the anterior end of the larvae. In a wide search of the literature, the authors of the present study only found little information about changes observed in *H. contortus* infective larvae exposed to FCF of NF. In another study, an *A. musiformis* strain was cultured in a modified CDB for 14 days at room temperature; the FCF was obtained to assess its in vitro nematocidal activity against *H. contortus* infective larvae. Interesting changes were recorded at the anterior end and also at the posterior end of larvae exposed to this FCF. A photographic analysis of *H. contortus* infective larvae exposed species to the FCF. This study showed a widening of larvae at the anterior end with the rupture of the cuticle. These findings were supported by an enzyme assay, where fungal protein was purified in a 9% polyacrylamide gel, co-polymerised with 1% gelatine (GS-PAGE), and finally, protease activity was identified [19]. The morphological changes observed in *H. contortus* infective larvae in our study could be attributed to a degrading process of cuticular tissues of perhaps to bioactive compounds, i.e., secondary metabolites with nematocidal activity produced by *A. musiformis*; however, this hypothesis should be proven through chromatographic techniques and perhaps by nuclear magnetic resonance imaging to determine the metabolite or metabolites responsible for the nematocidal activity. 

### 3.6. Mycochemical Compound Group Identification

The analysis of the mycochemical profile of *A. musiformis* growth in two different liquid media showed interesting results; for example, although both FCF showed the presence of alkaloids and saponins, a positive reaction (++) was only observed in sweet potato dextrose broth, while only the presence (+) of these compounds was observed in CDB. These results suggest that nutritional components in the culture medium influence the fungal production of extracellular secondary metabolites with an important nematocidal activity [54]. Sweet potato dextrose broth is an important source of sugars, fibre, lipids, vitamins, minerals and amino acids [55], while CDB provides only sugars and minerals (Sigma-Aldrich, Darmstadt, Germany). In this context, the more complete nutritive diet supplied by SPDB could influence a higher production of alkaloids and saponins than CDB. On the other hand, the mycochemical compound group analysis revealed the presence of coumarins only in liquid culture filtrates of *A. musiformis* cultured in CDB, and not in SPDB. Coumarins are phenolic compounds that are commonly found in some groups of plants and have been found to have important medicinal properties, including anti-fungal [56], antibacterial [57], and anti-tumour activities [58]. In a recent publication, two natural products identified as coumarin derivatives obtained from the plant *Ruta chalepensis* have been described for their anticancer, antidiabetic, antifertility, antimicrobial, antiplatelet aggregation, antiprotozoal, antiviral, and calcium antagonistic properties [59]. Similarly, coumarins from plants such as *Gliricidia sepium* (Fabaceae) and *Ruta chalepensis* (Rutaceae) have also been found to show nematocidal activity against gastrointestinal parasitic nematodes, i.e., *Cooperia punctata* and *H. contortus* affecting ruminants [60,61]. The phyto-chemical profile obtained in the present study suggests that coumarins produced in CDB and not in SPDB could be responsible for the nematocidal activity of the FCF of *A. musiformis**,* since FCF obtained from *A. musiformis* growing in SPDB did not show important nematocidal activity; in contrast, FCF of *A. musiformis* growth in CDB resulted in the highest nematocidal activity of >93%. This hypothesis should be proven through the separation of compounds by chromatographic techniques to determine the compound responsible for this activity. 

## 4. Materials and Methods

### 4.1. Allocation

This study was performed at the Laboratory of Helminthology, National Centre for Disciplinary Research in Animal Health and Innocuity from the National Institute of Research in Forestry, Agriculture and Livestock (INIFAP, SAGAR), at Jiutepec, Morelos State, Mexico. 

### 4.2. Fungal Isolation 

Soil samples were collected from a home garden in the Cuautla Municipality in the State of Morelos, Mexico. A small amount of soil (0.5 g) was sprinkled on water agar plates and some drops of a non-quantified amount of the free-living nematodes *Panagrellus redivivus* were added to the plate surface to act as “bites” to stimulate the predatory fungal structures of capture. Following a two-week incubation at room temperature (25–28 °C), the surfaces of the plates were reviewed under a light microscope, searching for the presence of conidiophore and conidia, the formation of trapping devices and trapped nematodes, among other characteristics. Once unique apical conidia conidiophores, similar to *A**. musiformis,* were identified, the monoconidial transfer to sterile water agar plates was carried out [62]. This process was repeated several times until the fungal isolates were contamination-free [41]. 

### 4.3. Traditional Morphological Taxonomy of Fungi

Traditional identification was carried out by observation and measurement of the most important taxonomic fungal structures such as conidia, conidiophores, septum and candelabrum, the presence of chlamydospores and the type of traps. For this aim, a total of 25 conidia and conidiophores were measured using a light microscope (20× and 40× objective lenses). Specialised taxonomic keys were used to determine the genera and species of the fungus [21,63].

### 4.4. Molecular Taxonomy of Fungi

DNA extraction from *A. musiformis* was performed using the Wizard^®^ Genomic DNA Purification Kit (Promega, Madison, WI, USA). The quantification was carried out using a IMPLEN spectrophotometer (NanoPhotometer NP80). Then, the DNA was amplified by PCR using the ITS4 (5′-GGAAGTAAAAGTCGTAACAAGG-3′) and ITS5 (5′-TCCTCCGCTTATTGATATGC-3′) primers [64], with a C1000 Touch^®^ Thermal Cycler (Bio-Rad, Hercules, CA, USA). PCR conditions were as follows: initial denaturation at 94 °C for 3 min; 35 cycles of denaturation at 94 °C for 1 min, annealing at 42 °C for 90 s and extension at 72 °C for 90 s; followed by a final extension stage at 72 °C for 5 min. Agarose gel electrophoresis with a 1.5% gel was used to confirm the size of amplicons, with products purified using the QIAquick gel extraction kit (QIAGEN) according to the manufacturer’s instructions. The samples were sequenced in the Institute of Biotechnology of the National Autonomous University of Mexico (IBT-UNAM) using an applied biosystem sequencer. The obtained sequences were aligned in NCBI-Blast www.ncbi.nlm.nih.gov/blast/’s accessed on 23 May 2022 (Basic Alignment Search Tool).

### 4.5. Nematodes 

#### 4.5.1. Panagrellus Redivivus

A strain of the free-living nematode *Panagrellus redivivus* was cultivated in crystal containers (10 cm width × 20 cm height) using oat grains and water following the procedure described by de Lara et al. (2007) [65]. Nematodes were recovered from cultures using the Baermann funnel technique and passed through 74 μm sieves to separate oat residues; finally, nematodes were rinsed in distilled water.

#### 4.5.2. *Haemonchus contortus* Infective Larvae

A population of *H. contortus* infective larvae was obtained from the faeces of an egg-donor lamb artificially infected with the parasite and maintained under confinement in pens in the flock experimental area of INIFAP. Faeces containing eggs of the parasite were directly collected from the rectum of the animal. Faecal material was used to elaborate coprocultures [66] and incubated for 7 days until the third larval stage was obtained. Larvae were recovered using a Baermann funnel system. The recovered larvae were washed by differential centrifugation using 40% sucrose for 3–5 min. Larvae were rinsed with tap water several times to eliminate the sucrose residues. Then, larvae were unsheathed with 0.187% sodium hypochlorite solution, where larvae remained within the range of 3–5 min and washed again to discard sheaths [67]. Clean larvae were resuspended in sterile distilled water and immediately used to perform the in vitro larval mortality assay.

### 4.6. Predatory Activity Assessment

One water agar cylinder (0.5 cm diameter × 0.3 cm height) from 15-day-old *A. musiformis* culture was transferred to 35 mm diameter water agar plates (n = 10) and incubated at room temperature (25–28 °C) for 7 days. Another set of 10 plates with only water agar medium was also included in the experiment to act as a control group. A hundred microliters of an aqueous suspension containing approximately two hundred *H. contortus* infective larvae were deposited on the surface of each plate. The whole plates were incubated at the same temperature for 7 days. After incubation, the agar from each plate was recovered and put on a Baermann funnel system in order to recover the whole non-trapped larvae [37]. The same procedure was used for the control agar plates. After 24 h on the funnel, larvae sedimented on the base of the assay tubes were quantified according to Olmedo-Juárez et al. (2022) [68]. The principle of larvae population reduction attributed to the fungal predatory activity was based on the mean of recovered larvae from the control group as 100% larvae without the fungal effect. Then, a comparison between both groups using the Abbott formula was used to estimate the larval reduction percentage as follows:

Abbot’s formula
$$\mathrm{PA}\;\%\;=\;\frac{\mathrm{RLc}\;-\;\mathrm{RLt}}{\mathrm{RLc}}\;*100$$
where PA% denotes the predatory activity (percentage), RLc indicates the mean of recovered larvae in plates with no fungi, and RLt shows the mean of recovered larvae in plates with fungi.

### 4.7. Nematocidal Activity of A. musiformis Liquid Culture Filtrates 

The *A. musiformis* strain was cultured on potato dextrose agar (PDA) plates at room temperature (25–28 °C) for two weeks. After incubation, three agar cylinders (0.5 cm height × 1 cm width) were taken from the FCF in PDA plates and were put into 250 mL Erlenmeyer flasks with 100 mL of CDB or SPDB-Am (*n* = 3). Controls without fungi containing only CDB-without fungus (WF) or SPDB-WF were included to discard any possible contamination. After incubation, the FCF with the growing fungus was filtered using 4 different filters as follows: (1) coffee filter paper; (2) Whatman paper no. 4; (3) syringe micro-filter 1.1μm; and (4) syringe micro-filter 0.22 μm. This procedure allows a sterile FCF to be achieved [69]. The resulting material was concentrated using a rotatory evaporator (45–50 °C, Büchi R-300, Flawil, Switzerland) and was totally dried using the lyophilisation process (Labconco, Kansas, MO, USA). The dried FCF was finally re-constituted by adding PBS (pH 7.2) according to the required concentration. The FCF/nematode confrontation was carried out using 96-well microtitre plates. Fifty microlitres of FCF was placed in each well (*n* = 3) and 50 μL of an aqueous suspension containing 100 *H. contortus* infective larvae was also placed in each well. The negative controls (*n* = 3) were: (1) PBS 7.2 pH; (2) CDB (dried) medium, re-constituted in PBS pH 7.2; and (3) SPDB dried medium, re-constituted in PBS pH 7.2. The following three FCF concentrations were assessed: 100, 50 and 25 mg/mL (Table 7). Readings were taken at 48 h post-treatment. Motionless larvae and larvae in movement were observed and quantified under the microscope at 5× and 10× magnification. Criterion to decide if motionless larvae were dead was achieved by touching their cuticle with a metallic needle to see if they responded with movements or remained motionless; larvae remaining motionless after this physical stimulus were considered as dead larvae. The means of motionless larvae and larvae in movement were recorded and compared between the experimental groups. The FCF that resulted in the highest lethal activity was analysed to identify the chemical groups of myco-constituents by myco-qualitative reagent analysis. 

### 4.8. Statistical Analyses

#### 4.8.1. Predation Assay

The predatory activity was statistically analysed using the Student t-test comparing the means of recovered larvae from the larvae/fungi confrontation plates in treated and control plates. 

#### 4.8.2. Fungal Culture Filtrate Assay

The results of the larval mortality attributed to the FCF effect were analysed using an ANOVA analysis, where the mean of dead and alive larvae were the response variables. The Tukey complementary test was used to identify at least one different mean with respect to the others.

### 4.9. Microscopic Analysis

A set of photographs illustrating the major taxonomic structures of the NF isolated, as well as *H. contortus* infective larvae captured by *A. musiformis* and dead larvae after exposure to FCF, were taken using a Leica DM6 B compound microscope in order to have documented proof of the fungal activity. 

### 4.10. Myco-Qualitative Reagent Analysis

The chemical profile was carried out using standard phytochemical test procedures with proper reagents and methods. Alkaloids were determined using the Dragendorff, Mayer and Wagner’s reagents. The presence of coumarins was determined by the Bornträger test, while the Mg^2+^ and HCl tests were used for flavonoids. The ferric chloride, gelatine and saline solution tests were used for tannins. Triterpenes were determined using the Lieberman–Burchard and Salkowski tests [68].

## 5. Conclusions

The results of the present study led us to conclude that the NF *A. musiformis* cultured in Czapek–Dox broth medium produces mycochemical constituents that are verted into medium, and its FCF exert an important in vitro nematocidal activity against *H. contortus* infective larvae. Similarly, coumarins were the mycochemical group of compounds identified in FCF obtained from *A. musiformis* growth in CDB that exhibited an important nematocidal activity; meanwhile, this compound was not identified in the non-active FCF obtained from *A. musiformis* growth in sweet potato dextrose broth medium. Liquid culture filtrates of *A. musiformis* growth in CDB should be explored through chromatographic and nuclear magnetic resonance procedures to determine the compound responsible for the nematocidal activity.

## Figures and Tables

**Figure 1 pathogens-11-01068-f001:**
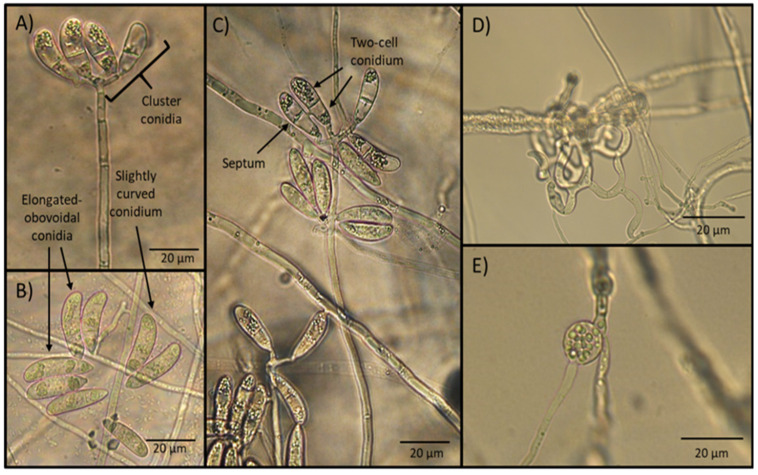
Microphotographs showing the aspect of taxonomic characteristics (conidiophores and conidia), typical of the nematophagous fungus *Arthrobotrys*
*musiformis*. (**A**) Erect conidiophore crowned with conidia clusters; (**B**) conidia present elongated–obovoidal shape and are slightly curved; (**C**) conidia are formed with two cells separated by a septum; (**D**) three dimensional adhesive nets; (**E**) one developing chlamydospore.

**Figure 2 pathogens-11-01068-f002:**
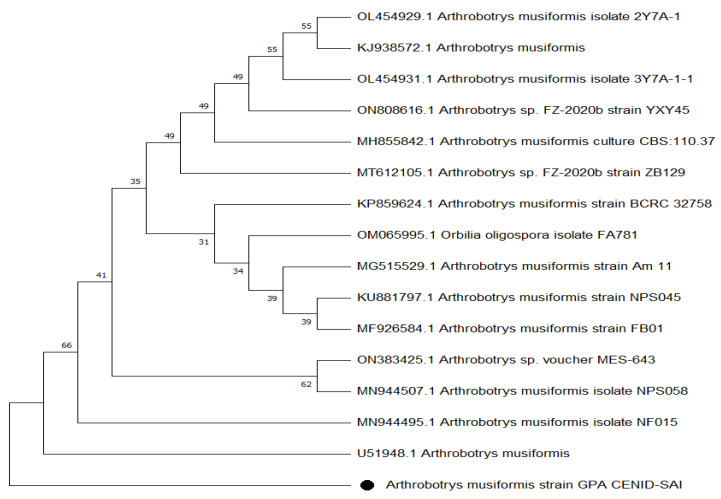
Phylogenetic tree of *Arthrobotrys*
*musiformis*.

**Figure 3 pathogens-11-01068-f003:**
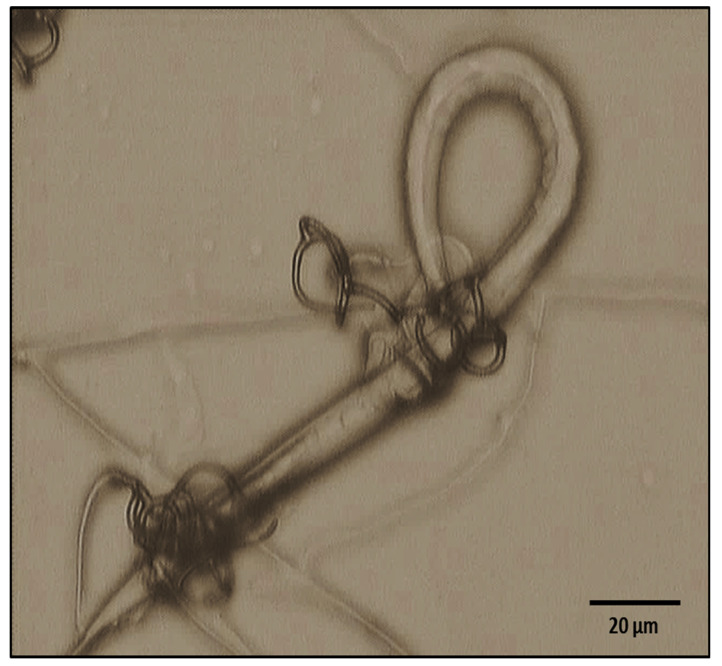
Photograph showing a *Haemonchus*
*contortus* infective larvae captured into a three-dimensional adhesive net of *Arthrobotrys*
*musiformis* after 7 days of interaction.

**Figure 4 pathogens-11-01068-f004:**
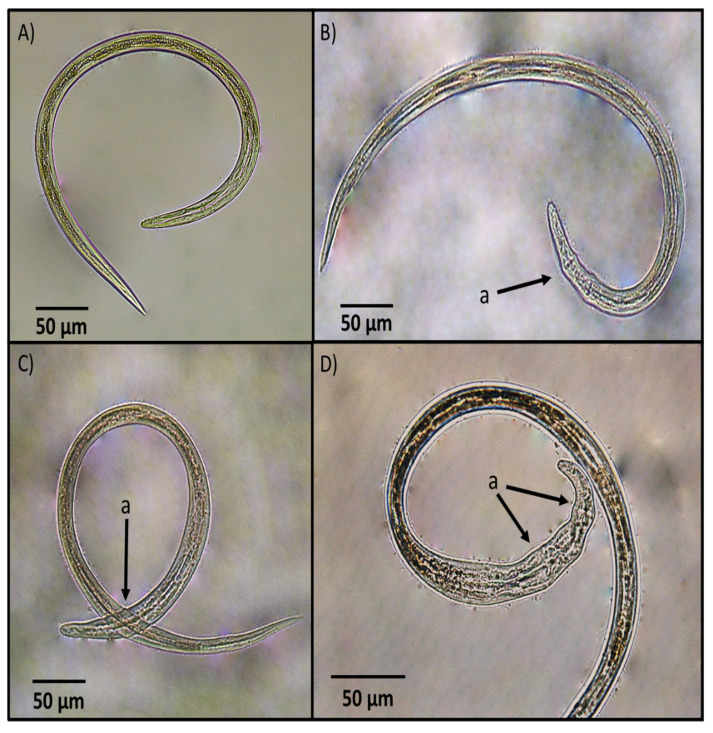
Comparative photographs showing: (**A**) an unsheathed *Haemonchus*
*contortus* infective larva from the control group; (**B**–**D**) dead unsheathed larvae after exposure to *Arthrobotrys musiformis* liquid culture filtrate showing morphological malformations (marked with an arrow as a) mainly at the anterior extreme.

**Table 1 pathogens-11-01068-t001:** Mean and range of 25 conidia and conidiophores measurements, and characteristics observed under a light microscope.

Characteristic	Mean (µm)	Range (µm)
Conidia length	36.16	30.11–40.08
Conidia width	8.99	7.66–10.29
Conidiophore length	240.8	166–407
Chlamydospores	Present	
Type of traps	Adhesive nets	

**Table 2 pathogens-11-01068-t002:** Similarity and coverage of the obtained sequence after comparison with reported sequences by the GenBank–NCBI base, using the partial sequence ITS1, 5.8S and ITS2 regions.

Strain	Query Cover %	Similarity %	Gen Bank Accession Number
*Arthrobotrys* sp. FZ-2020b	96	99.83	MT612105.1
*A. musiformis* CBS 110.37	96	99.83	MH855842.1
*A. musiformis* 3Y7A-1–1	96	99.83	OL454931.1
*A. musiformis* Am_11	96	99.66	MG515529.1
*A. musiformis*	96	99.66	KJ938572.1

**Table 3 pathogens-11-01068-t003:** Mean of *Haemonchus*
*contortus* infective larvae recovered from *Arthrobotrys*
*musiformis* water agar plates after 7 days of confrontation. *p* < 0.01.

Group	Mean of Recovered Larvae ± SD	Reduction Larvae %
Group 1 Larvae/Fungus Interaction	62 ± 41.67	71.54%
Group 2 Larvae (Control)	217 ± 28.05	

**Table 4 pathogens-11-01068-t004:** Mean of dead and total *Haemonchus*
*contortus* infective larvae recovered from the wells of microtitre plates after 48-hour interaction with *Arthrobotrys*
*musiformis* culture filtrates and mortality percentages.

Group	100 mg/mL		50 mg/mL		25 mg/mL	
	Dead/Total	Mortality %	Dead/Total	Mortality %	Dead/Total	Mortality %
CzDox- *A. musiformis*	99/106	93.42 ± 10.49 a	75/103	73.02 ± 16.02 a	49/95	51.61 ± 19.41 a
SPDB- *A. musiformis*	26/97	26.80 ± 2.76 b	18/97	18.42 ± 8.98 b	15/87	16.91 ± 2.37 b
CzDox- No Fungus	4/98	4.45 ± 2.55 c	3/98	3.48 ± 2.88 b	5/74	6.76 ± 4.77 b
SPDB- No Fungus	5/83	5.51 ± 3.97 c	8/96	8.77 ± 2.01 b	9/108	8.38 ± 1.07 b
PBS	6/98	6.00 ± 2.29 c	6/98	6.00 ± 2.29 b	6/98	6.00 ± 2.29 b

CzDox = Czapek–Dox broth; SPDB = sweet potato dextrose broth; PBS = phosphate buffer solution; means with different letters show statistical significance; *p* < 0.05; *n* = 3.

**Table 5 pathogens-11-01068-t005:** Myco-constituent groups of *A. musiformis* culture filtrates.

Metabolite Group	Assay	Colourimetric Reaction	Fungus in Czapek–Dox Broth	Fungus in Sweet Potato Dextrose Broth
Alkaloids	Dragendorff	Change of colour to brown	+	++
	Mayer	Change of colour to yellow	+	++
	Wagner	and formation of precipitate	+	++
* Coumarins	Bornträger	Yellow fluorescence after 24 h (see in U.V)	+	−
Flavonoids	Mg2 + y HCL	Red, orange and violet colours	−	−
Tannins	Ferric chloride	Hydrolysable tannins (blue)	−	−
		Condensed tannins (green)	−	−
	Confirmation	White precipitate	−	−
	Solution of gelatin			
	Gelatin and saline solution			
	Saline solution			
Triterpenes/Sterols	Reaction of Liebermann- Buchard	Blue or blue-green (sterols)	−	−
	Reaction of Salkowski	Red or purple (triterpenes)	−	−
Saponins	Foam formation	Persistent foam formation	+	++

− absence; + presence; ++ positive reaction; * coumarins present only in CDB.

**Table 6 pathogens-11-01068-t006:** Results about the in vitro predatory activity of *Arthrobotrys*
*musiformis* isolates against different blank nematodes.

Blank Nematode	Host	Predatory Activity %	Authors
*Scutellonema bradys*	Yam tubers	94.6%	[36]
*H. contortus*	Sheep	97%	[37]
*H. contortus*	Small ruminants	90.4%	[38]
*Trichostrongylidae*	Ruminants	60.72–99.95%	[39]
*Trichostrongylus colubriformis*	Ruminants	94.8%	[17]
*H. contortus*	Sheep	100%	[40]
*Panagrellus redivivus*	A non-parasitic free-living nematode	62.7–93.6%	[41]
*Meloidogyne hapla*	Tomatoes	97%	[42]

**Table 7 pathogens-11-01068-t007:** Experimental design.

Group	Medium	Fungus
Group 1	Czapek–Dox broth	*A. musiformis*
Group 2	Sweet potato dextrose broth	*A. musiformis*
Group 3	Czapek–Dox broth	No fungus *
Group 4	Sweet potato dextrose broth	No fungus *
Group 5	PBS control	No fungus *

*n* = 3 wells/plate; Readings: 48 h post-treatment; * = negative controls; three assessed concentrations: 25, 50 and 100 mg/mL.

## Data Availability

Not applicable.

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
