# Peer review of "Arthrobotrys musiformis (Orbiliales) Kills Haemonchus contortus Infective Larvae (Trichostronylidae) through Its Predatory Activity and Its Fungal Culture Filtrates"

_pathogens, 2022, doi:10.3390/pathogens11101068_

Round 1

Reviewer 1 Report

There are not sufficient originalitty or  interest to the readers, despite the work being well written.

Author Response

Reviewer 1

Comments and Suggestions for Authors

The paper is well-written and carried out with adequate methodology, however, the

authors could at least have evaluated the nematicidal effect of A. musiformis against

infective larvae of H. contortus (L3) in grass pots. There are so many paper published

with this issue, like:

Vet Rec Open. 2015; 2(1): e000103.

Published online 2015 May 16. doi: 10.1136/vetreco-2014-000103

Proteolytic activity of extracellular products from Arthrobotrys musiformis and their effect

in vitro against Haemonchus contortus infective larvae

Even from 90”s:

Biological control "in vitro" of infective Haemonchus placei larvae by predacious fungi

Arthrobotrys musiformis. Araújo, M. J. V; Santos, M. A; Ferraz, S; Maia, A. S.Arq. bras.

med. vet. zootec ; 46(3): 197-204, jun. 1994.

Despite being a good work, there is no recommendation for publication in this journal due

to the lack of originality of the article.

There are not sufficient originalitty or  interest to the readers, despite the work being well written.

Authors response:

Dear Reviewer,

There is quite a lot of information about predatory activity and fungal culture filtrates with nematocidal activity with Arthrobotrys oligospora and other species; however, in an extensive review of literature available on A. musiformis we found a really limited information and we think this study open new expectations to keep ahead with exploring the nematocidal activity of culture filtrates of this species and its secondary metabolites with nematocidal activity that could lead to obtaining new natural anthelmintics.

Reviewer 2 Report

Congratulations to the authors; this is a well written manuscript, methodologically very well detailed and with significant findings that should be reported. Only few observations are recommended.

Line 121. Include a brief description from table 3.

Line 400. Establish the criteria for larval death

References. Check that all references are uniform and under the guidelines of the journal. Special care with the use of capital letters.

Author Response

Reviewer 2

Comments and Suggestions for Authors

Congratulations to the authors; this is a well written manuscript, methodologically very well detailed and with significant findings that should be reported. Only few observations are recommended.

Line 121. Include a brief description from table 3.

Line 400. Establish the criteria for larval death

References. Check that all references are uniform and under the guidelines of the journal. Special care with the use of capital letters.

Authors response:

Dear Reviewer,

We greatly appreciate your kind comments.

We have prepared a new manuscript considering your comments and suggestions, next:

 Line 121. Include a brief description from table 3.

Authors: We have including a brief description from Table 3

Line 400. Establish the criteria for larval death

Authors: In the new version of our manuscript we have including the criterion to establish if larvae were dead or live.

References. Check that all references are uniform and under the guidelines of the journal. Special care with the use of capital letters.

Authors: We have checked all the references and we have corrected all errors we found according to the guidelines of the journal.

Round 2

Reviewer 1 Report

Adjustments made as requested. The work is not innovative, but it was properly executed and well written.